# Safety and Feasibility Assessment of a Pharmacy-Driven AUC/MIC Vancomycin Dosing Protocol in a Multicenter Hospital System

Morgan Phillips,[a] Jeremy Rose,[a] Ashlee Hamel,[b] Amanda Ingemi[b]

aSentara RMH Medical Center, Harrisonburg, Virginia, USA
bSentara Norfolk General Hospital, Norfolk, Virginia, USA

**ABSTRACT**    Vancomycin is used for Gram-positive infections, including methicillin-resistant *Staphylococcus aureus*. The 2020 vancomycin guidelines described by M. J. Rybak, J. Le, T. P. Lodise, D. P. Levine, et al. (Am J Health Syst Pharm 77:835–864, 2020, https://doi.org/10.1093/ajhp/zxaa036) provided an update on vancomycin dosing, which recommended an optimal area under the concentration-time curve over 24 h to MIC (AUC/MIC) target of 400 to 600. In 2021, a pharmacy-driven AUC/MIC vancomycin dosing protocol was implemented across 12 Sentara Health System hospitals. The primary objective of this study was to assess if the pharmacy-driven AUC/MIC vancomycin dosing protocol led to fewer acute kidney injury (AKI) events than trough-based dosing. Secondary objectives included vancomycin duration, hospital length of stay, administered vancomycin dose during admission, vancomycin labs drawn during standard lab times, and cost. AKI was assessed in two separate ways: (i) modified AKIN (Acute Kidney Injury Network) criteria and (ii) a modified version from the vancomycin guidelines. Inferential statistics were used to analyze the results of this retrospective study. Per the AKIN definition, the rates of AKI were 13.9% (349/2,507) in the trough-based group and 14.9% (369/2,471) in the AUC/MIC-based group ($P = 0.309$). Per the definition of the vancomycin guidelines, the rates of AKI were 6.7% (169/2,507) in the trough-based group and 7.6% (187/2,471) in the AUC/MIC-based group ($P = 0.258$). A total of 52% (2,679/5,151) of vancomycin labs were obtained during standard lab times in the AUC group and 24% (1,144/4,766) in the trough group ($P < 0.05$). There was no difference in AKI events between AUC and trough dosing. Use of contrast dye may confound these results. AUC/MIC dosing was associated with more lab draws during standard times, a larger number of labs drawn per person, and less total use of vancomycin.

**IMPORTANCE**    In this article, we report that there were no differences in rates of acute kidney injury between trough-based vancomycin dosing and AUC/MIC-based vancomycin dosing across 12 hospitals. AUC/MIC dosing resulted in more vancomycin lab draws during standard lab draw times compared to trough dosing, thus making it more convenient for health care personnel. This study includes all uses for vancomycin, including empirical use, and all patient severity levels. Therefore, this research reflects real-world use of vancomycin in hospitals. AUC/MIC dosing is supported by various infectious disease societies. However, the feasibility of incorporating AUC/MIC dosing in hospitals is undetermined. This study is unique in that it includes hospitals of various sizes (small community hospitals and an academic teaching hospital), and it includes a feasibility component. Therefore, this study has broad applicability to other hospitals across the United States. This original research includes the clinical application of vancomycin in a multicenter health system.

**KEYWORDS**   Gram-positive bacteria, pharmacokinetics, vancomycin

Address correspondence to Morgan Phillips, morgan.phillips973@gmail.com.

The authors declare no conflict of interest.

**V**ancomycin is used to treat bacterial infections caused by Gram-positive organisms, including methicillin-resistant *Staphylococcus aureus* (MRSA) and *Enterococcus faecalis* (1). It is also used empirically to cover MRSA until culture information is available.

Vancomycin has notable side effects of which nephrotoxicity is often most clinically relevant (2). Given its significant potential for toxicity, vancomycin is considered a narrow therapeutic drug, and drug concentrations must be monitored throughout therapy. Prior to 2009, the consensus was to utilize peak and trough concentrations for therapeutic monitoring. However, recommendations per the 2009 vancomycin dosing guidelines reflected a vancomycin target area under the concentration-time curve over 24 h to MIC (AUC/MIC) of ≥400 (3). Given the limited software available to many institutions at the time, a trough concentration of 10 to 20 mg/L was recommended as an appropriate surrogate marker for achieving clinical cure in patients with stable renal function.

Despite this recommendation, there was a concern that targeting higher trough levels would result in unnecessary nephrotoxicity. A single-center, quasi-experiment with 1,280 hospitalized adult patients showed that AUC-guided dosing was independently associated with lower rates of vancomycin-induced nephrotoxicity (logistic regression; odds ratio [OR], 0.52; 95% confidence interval [CI], 0.34 to 0.80; Cox proportional hazards regression [HR], 0.53; 95% CI, 0.35 to 0.78) (4).

To address correlation of trough-based dosing to AUC, a study by Neely et al. published in 2014 found that trough-only and trough-peak data sets in 47 adults underestimated the true AUC compared to a full model (Pmetrics nonparametric population modeling package) (5). However, when using Bayesian modeling with trough-only data, it allowed 97% accurate AUC estimation (93% to 102%; $P = 0.23$). Additional studies have highlighted the effective utility of estimating AUC/MIC through Bayesian modeling techniques. One caveat to note is that most of these studies were performed in patients with MRSA bacteremia, with few looking at pneumonia and infective endocarditis.

Cumulatively, these studies helped to form the updated 2019 vancomycin dosing guidelines for therapeutic monitoring for serious MRSA infectious. The guidelines suggest an individualized target AUC/MIC of 400 to 600 (assuming a vancomycin MIC of 1 mg/L), with target exposure achievement during the first 24 to 48 h (6). Bayesian-derived AUC monitoring is a judicious use of therapeutic vancomycin monitoring as steady-state concentrations are not required.

Sentara Health Systems was able to implement a pharmacy-driven vancomycin protocol utilizing AUC/MIC Bayesian modeling software, InsightRx, which is integrated into Epic's electronic health system. An advantage to an integrated software allows for modeling of an AUC/MIC-based vancomycin regimen that incorporates patient-specific factors. Of note, while the 2019 guidelines suggest this target for patients with serious MRSA infections, Sentara Health Systems has implemented AUC/MIC dosing for all indications. This includes nonserious infections and indications where MRSA may not be suspected but is being utilized for empirical coverage. Finally, trough-based dosing is utilized for patients with unstable renal function.

The goal of this research project is to assess the safety and feasibility of a new AUC/MIC pharmacy-driven protocol compared to previous trough-based vancomycin therapeutic drug monitoring. Results of this study will augment the existing literature by including a real-world application of a standardized approach to AUC/MIC vancomycin dosing in a hospital system for a variety of infectious disease diagnoses.

## RESULTS

A total of 2,507 patients were included in the trough-based group, and 2,471 patients were included in the AUC/MIC-based group (Fig. 1). Baseline characteristics are represented in Table 1. There were 237/2,507 (9.4%) ICU admissions in the trough group compared to 187/2,471 (7.6%) in the AUC/MIC group ($P < 0.05$). Nephrotoxin drug use was not significantly different between the two groups: 2,013/2,507 (80.3%) in the trough group compared to 1,979/2,471 (80.1%) in the AUC group ($P = 0.9$). There

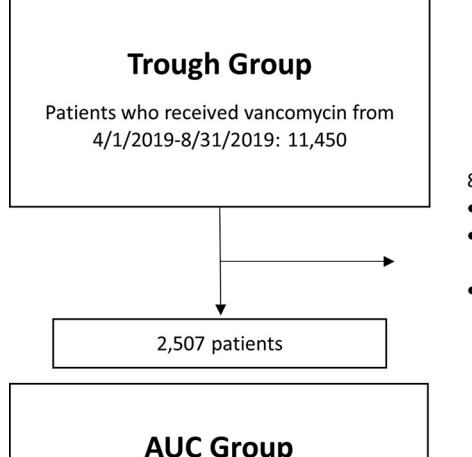

**Trough Group**

Patients who received vancomycin from 4/1/2019-8/31/2019: 11,450

8,943 patients excluded
- 419 for no vancomycin level ordered
- 3,887 for no dosing strategy, dialysis, unstable renal function
- 4,637 for hospital duration < 4848 hours, vancomycin < 48 hours

2,507 patients

**AUC Group**

Patients who received vancomycin from 4/1/2021-8/31/2021: 9,703

7,232 patients excluded
- 321 for no vancomycin level ordered
- 1,666 for no dosing strategy, dialysis, unstable renal function
- 5,245 for hospital duration < 48 hours, vancomycin < 48 hours

2,471 patients

**FIG 1** The exclusion criteria of this study were as follows: vancomycin duration of <48 h, hospital length of stay of <48 h, no available dosing strategy, patients on any type of dialysis, unstable renal function (as clinically determined and documented by the clinical pharmacist or a baseline serum creatinine of >1.9 mg/dL), and no vancomycin level ordered. There were 2,507 patients in the trough group and 2,471 patients in the AUC group.

was less contrast dye use in the trough group (1,214/2,507 [48.4%]) than in the AUC/MIC group (1,476/2,471 [59.7%]) ($P < 0.05$). The mean initial serum creatinine on admission was 0.89 mg/dL for both groups.

There was no difference in acute kidney injury (AKI) events between the two groups (Table 2), regardless of definition use.

Results for secondary endpoints are shown in Table 3. A total of 52% (2,679/5,151) of patients in the AUC group had vancomycin laboratory draws within standard nursing blood draw times compared to 24% (1,144/4,766) in the trough group ($P < 0.05$). The vancomycin duration was 3.5 days for the trough group versus 3.6 days for the AUC/MIC group ($P < 0.05$). The total vancomycin dose per vancomycin patient-day was greater in the trough group (2,643 mg) than in the AUC/MIC group (2,292 mg) ($P < 0.05$). The costs of vancomycin per person, based on the median dose per vancomycin patient-day and vancomycin labs per vancomycin patient-day, were $66.9 for the trough group and $71.7 for the AUC group ($P < 0.05$). On average, 0.58 vancomycin labs

**TABLE 1** Baseline characteristics

| Characteristic | Result for group[a] | | P value |
| --- | --- | --- | --- |
| | Trough (n = 2,507) | AUC (n = 2,471) | |
| ICU admission, count (%) | 237 (9.4) | 187 (7.6) | <0.05 |
| Nephrotoxin use for ≥24 h, count (%) | 2,013 (80.3) | 1,979 (80.1) | 0.9 |
| Received contrast dye, count (%) | 1,214 (48.4) | 1,476 (59.7) | <0.05 |
| Age, mean ± SD yr | 62.57 ± 16.69 | 63.14 ± 16.29 | 0.223 |
| Actual body wt, mean ± SD kg | 88 ± 28 (n = 2,445) | 89 ± 30 (n = 2,379) | 0.338 |
| Baseline serum creatinine, mean ± SD mg/dL | 0.89 ± 0.37 | 0.89 ± 0.35 | 0.473 |

[a]The trough-based group included patients' samples collected from 1 April 2019 through 31 August 2019. The AUC-based group included patients' samples collected from 1 April 2021 through 31 August 2021.

**TABLE 2** Primary safety endpoints of rate of AKI

| AKI[a] | Count (%) for group | | P value |
| | Trough (n = 2,507) | AUC (n = 2,471) | |
|---|---|---|---|
| Definition 1 | 349 (13.9) | 369 (14.9) | 0.309 |
| Definition 2 | 169 (6.7) | 187 (7.6) | 0.258 |

[a]AKI, acute kidney injury. Definition 1 is based on the AKIN criteria: an increase in serum creatinine of ≥0.3 mg/dL or ≥50% on two consecutive measurements. Definition 2 is based on the vancomycin guidelines: an increase in serum creatinine of ≥0.5 mg/dL or ≥50% (whichever is greater) on two consecutive measurements.

were drawn per vancomycin patient-day in the AUC/MIC group compared to 0.54 per vancomycin patient-day in the trough group ($P < 0.05$). Hospital length of stay was longer in the AUC group (10.8 days) compared to the trough group (9.3 days) ($P < 0.05$).

## DISCUSSION

This retrospective study sought to compare the safety and feasibility of a pharmacy-driven AUC/MIC vancomycin protocol to previous trough-based dosing. There was no significant difference in the primary endpoints of rate of AKI using two definitions of AKI. This result was unexpected as prior literature has shown reductions in nephrotoxicity rates with AUC/MIC vancomycin dosing compared to trough-based dosing. There are factors that may explain these unexpected results and that may have led to an underrepresentation of the true impact of AUC/MIC dosing at Sentara hospitals. A greater number of patients in the AUC group had intravenous (i.v.) contrast dye use. Intravenous contrast dye is known to increase the risk of AKI in hospitalized patients (7). There was no identifiable reason for higher contrast dye use in the AUC group. This could be interrogated in future studies of the AUC protocol. Another important factor to consider is the median vancomycin duration in both groups was ~3.5 days and was skewed to shorter durations. In most vancomycin-induced nephrotoxicity cases, the incidence occurs after at least 4 days of therapy on vancomycin. In the trough-based group, there were significantly greater critical care admissions than in the AUC-based group. Critical care patients are more likely to exhibit unstable renal function and AKI. Therefore, the rate of AKI collected in this study may not reflect the true rate of vancomycin-induced AKI. Interestingly, hospital length of stay was longer in the AUC group, which could be explained by factors such as comorbidities and infection severity. This is in contrast to a short duration of vancomycin, possibly indicating other factors involved in hospital length of stay.

An advantage of AUC dosing was the reduced exposure to vancomycin even with a slightly longer duration of vancomycin therapy. From a cost perspective, this resulted in a difference of $2.70 per person. This reduced exposure is not surprising as the benefit of using AUC dosing is the ability to obtain a vancomycin level earlier in therapy than with trough-based dosing. Therefore, a pharmacist can appropriately adjust therapy earlier rather than waiting until steady-state conditions to obtain a trough level. Additionally, an AUC/MIC of 400 to 600 usually correlates to lower trough levels, which theoretically would result in fewer AKI events.

**TABLE 3** Secondary endpoints

| Characteristic | Result for[a]: | | P value |
| | Trough (n = 2,507) | AUC (n = 2,471) | |
|---|---|---|---|
| Vancomycin labs collected between 03:00 and 06:00, count (%) | 1,144 (24) | 2,679 (52) | <0.05 |
| Vancomycin duration, median days (IQR) | 3.5 (2.6) | 3.6 (2.9) | <0.05 |
| Total vancomycin dose/vancomycin patient-day, median mg (IQR) | 2,643 (1,929) | 2,292 (1,597) | |
| Cost of vancomycin dose/person, median $ (IQR) | 24.7 (14) | 22 (15.4) | <0.05 |
| Hospital length of stay, mean ± SD days | 9.3 ± 11.2 | 10.8 ± 11.4 | <0.05 |
| Vancomycin labs/vancomycin patient-day, mean ± SD | 0.54 ± 0.4 | 0.58 ± 0.39 | |
| Cost of vancomycin labs/vancomycin patient-day, mean ± SD $ | 66.9 ± 47.7 | 71.7 ± 48.8 | |

[a]If data are nonparametric, results are reported as medians and interquartile ranges (IQRs). The cost of vancomycin is $2.67/g, and the cost of a vancomycin lab is $123.

Another advantage that this study revealed was nursing and pharmacy convenience. Laboratory blood draws are performed in the morning around 04:00 a.m. at Sentara hospitals. With the ability to collect a vancomycin lab at almost any time using AUC dosing, the pharmacist can more easily schedule the vancomycin level to be drawn at 04:00. This leads to fewer needle sticks for patients and less interruption of nursing workflow. Anecdotally, pharmacists prefer using the new AUC/MIC protocol as the integrated software improves efficiency, allows the pharmacist to spend more time on other important duties, and eases workload during lower staffing times (evening and night pharmacists). Despite the convenience component, there were slightly more lab draws collected per person in the AUC group compared to the trough group (2.1 labs per person versus 1.9 labs per person, respectively), and thus there was a cost difference of $24 per person. This was to be expected as the new AUC protocol allows a pharmacist to draw a vancomycin lab prior to steady state. Since trough-based dosing requires a level to be drawn at steady state and steady state can take several days to reach, vancomycin may be discontinued prior to a level being drawn. (The average vancomycin duration was only 3.5 days in this study.) In this study, the vancomycin durations tended to be shorter (prior to achievement of steady state). If this study looked at only vancomycin durations greater than 4 days, there may not be any difference in lab draws per person between the two groups.

A limitation of this study was that the AUC data were collected 4 months into the new protocol rollout. There is a learning curve to implementing a new workflow procedure, and therefore, inadequate pharmacist documentation was observed. We initially sought to collect infectious disease diagnoses at initiation of vancomycin to help differentiate between nonsevere and severe infections. The majority of the AUC group was lacking indication documentation, limiting the applicability of this parameter. If one group had more serious infections than the other group, it could have impacted the results and we would be unable to capture this in the study. As further experience with this new protocol and education are provided to hospital staff, we hope to collect additional data for future analysis.

Another limitation of this study was defining the primary endpoint of rate of AKI. Despite using definitions recommended by the guidelines, it is difficult to directly associate an AKI with vancomycin administration. This study aimed to account for as many factors that could contribute to AKI, but other factors could have influenced a patient's risk for AKI. Also, use of serum creatinine alone may not be the most accurate means for defining AKI. Additional measures, such as decrease in urine output or change in glomerular filtration rate, could improve confidence of accurately defining AKI. Given the large amount of patient information collected and the retrospective nature of this study, collecting these additional measures was not feasible.

This retrospective, multicenter study showed overall low rates of AKI using an AUC/MIC-based vancomycin dosing protocol compared to the literature. One article published by Elyasi et al. reviewed over 50 studies looking at rate of vancomycin-induced AKI and found rates from 10% to 40%, with higher rates being associated with high-dose regimens (8). The rate of AKI was not statistically different from a historical, trough-based cohort. AUC dosing led to improvements in nursing and pharmacy workflows and overall lower exposure to vancomycin. This is one of the first studies looking at safety and feasibility of a vancomycin AUC/MIC dosing protocol in a multicenter hospital system that includes 11 community hospitals and 1 teaching hospital. In the future, we hope to collect additional data after pharmacists gain more experience with the newly implemented protocol, and we hope to assess vancomycin utility in specific infectious disease diagnoses. Additionally, there is room for improvement in education for obtaining fewer vancomycin levels given our data showing the short duration of vancomycin use. While this study aimed to look at safety and feasibility, clinical outcomes was not included in our evaluation. In future studies, collection of in-hospital mortality, infection recurrence, etc., could be performed. In conclusion, this study showed that an integrated Bayesian modeling software for vancomycin AUC/MIC dosing is safe and feasible for use among hospitals of various sizes.

## MATERIALS AND METHODS

**Study design.** This was a multicenter, retrospective cohort study that took place at 12 Sentara Healthcare hospitals. 11 hospitals are community hospitals that range from 150 to 300 beds. One hospital is an academic hospital with 563 beds. Patient data were collected from the hospital EHR (electronic health record). Patients were included if they were 18 years and older, received intravenous vancomycin for ≥48 h, had at least 1 vancomycin level drawn, and had a hospital length of stay of ≥48 h. Excluded patients were those who underwent dialysis and had unstable renal function as determined by clinical pharmacist evaluation and documentation in the first 48 h of admission or if the baseline serum creatinine was ≥1.9 mg/dL.

Patient data for the trough-based dosing group were collected from all 12 hospitals from 1 April 2019 through 31 August 2019, and those for the AUC/MIC-based dosing group data were collected from 1 April 2021 through 31 August 2021.

This study design was approved by the local Sentara RMH Medical Center Institutional Review Board.

**Vancomycin AUC/MIC pharmacy dosing protocol.** At Sentara Healthcare, providers entered orders for pharmacy to dose vancomycin. Providers then entered indication and length of therapy. Outside of one-time surgical dosing, pharmacists managed all aspects of vancomycin dosing and monitoring without physician input. If a pharmacist wanted to discontinue vancomycin, they then reached out to the provider with that recommendation. Across the 12 sites, ~150 vancomycin consults were completed daily by 40 pharmacists.

Sentara Healthcare purchased a proprietary AUC/MIC Bayesian modeling software (InsightRx) for vancomycin and adjusted the pharmacy-driven protocol to reflect dosing per AUC/MIC. With this protocol update, pharmacist documentation and handoffs were changed. This updated protocol was approved by Pharmacy and Therapeutics and was officially implemented in January of 2021 at all Sentara hospitals. Pharmacist training took place during the rollout of the AUC/MIC software and updated pharmacy-driven protocol. The AUC/MIC software allowed pharmacists to obtain a random vancomycin level prior to steady-state conditions. In the past, the trough-based pharmacy-driven protocol recommended obtaining a trough level prior at steady-state conditions (prior to the 4th or 5th vancomycin dose).

**Study endpoints.** The primary endpoint of this study was rate of acute kidney injury (AKI). The primary endpoints were defined as described below.

**Primary endpoints. (i) AKIN criteria.** By the AKIN (Acute Kidney Injury Network) criteria (9), the primary endpoint was defined as an increase in serum creatinine of ≥0.3 mg/dL or ≥50%, on two consecutive measurements. This modified version excludes a urine volume of ≤0.5 mL/kg of body weight/h for 6 h in the definition as this measure was not a feasible parameter to collect with our data reporting software.

**(ii) Vancomycin guidelines.** By the vancomycin guidelines (10), the primary endpoint was defined as an increase in serum creatinine of ≥0.5 mg/dL or ≥50% (whichever is greater) on two consecutive measurements. This modified version excludes a decrease in calculated creatinine clearance of 50% from baseline on 2 consecutive days as this measure was not a feasible parameter to collect with our data reporting software.

**Secondary endpoints.** Secondary endpoints included hospital length of stay, cost (lab draws, vancomycin dose), duration of vancomycin therapy in hours, and number of vancomycin lab draws outside of standard morning lab draws (03:00 to 06:00).

**Baseline characteristics.** The baseline characteristics collected were age, weight, serum creatinine on admission, infectious disease diagnosis, concomitant nephrotoxic agents during hospital stay for ≥24 h, use of intravenous contrast dye, and intensive care unit (ICU) admissions. If the patient received: piperacillin-tazobactam, loop diuretic, nonsteroidal anti-inflammatory drugs, aminoglycoside, angiotensin-converting enzyme inhibitor, or angiotensin receptor blocker, they were considered to have received a concomitant nephrotoxic agent.

**Statistical analysis.** For continuous variables, an equal variance $t$ test for normally distributed data (reported as means and standard deviations [SD]) or Mann-Whitney U test for nonparametric data (reported as medians and interquartile ranges [IQRs]) was used. For nominal variables, a chi-square test was used.

## ACKNOWLEDGMENT

We declare no conflict of interest.

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
