## [Reviewer comments · Microbiology Spectrum]

Microbiology Spectrum

Safety and Feasibility Assessment of a Pharmacy-Driven AUC/MIC Vancomycin Dosing Protocol in a Multicenter Hospital System

Morgan Phillips, Jeremy Rose, Ashlee Hamel, and Amanda Ingemi

Corresponding Author(s): Morgan Phillips, Sentara RMH Medical Center

Review Timeline:

Submission Date:	August 26, 2022
Editorial Decision:	September 22, 2022
Revision Received:	December 19, 2022
Editorial Decision:	December 20, 2022
Revision Received:	December 21, 2022
Editorial Decision:	December 24, 2022
Revision Received:	January 13, 2023
Accepted:	January 19, 2023

Editor: Tulip Jhaveri

Reviewer(s): Disclosure of reviewer identity is with reference to reviewer comments included in decision letter(s). The following individuals involved in review of your submission have agreed to reveal their identity: Kayla R Stover (Reviewer #2)

Transaction Report:

DOI: <https://doi.org/10.1128/spectrum.03313-22>

September 22, 2022

Dr. Morgan Simonian Phillips
Sentara RMH Medical Center
2010 Health Campus Drive
Harrisonburg, Virginia 22801

Re: Spectrum03313-22 (Safety and Feasibility Assessment of a Pharmacy-Driven AUC/MIC Vancomycin Dosing Protocol in a Multicenter Hospital System)

Dear Dr. Morgan Simonian Phillips:

Thank you for submitting your manuscript to Microbiology Spectrum. I am willing to consider this manuscript, however major revisions are needed as outlined by the expert reviewers. When submitting the revised version of your paper, please provide (1) point-by-point responses to the issues raised by the reviewers as file type "Response to Reviewers," not in your cover letter, and (2) a PDF file that indicates the changes from the original submission (by highlighting or underlining the changes) as file type "Marked Up Manuscript - For Review Only". Please use this link to submit your revised manuscript - we strongly recommend that you submit your paper within the next 60 days or reach out to me. Detailed instructions on submitting your revised paper are below.

Link Not Available

Sincerely,

Tulip Jhaveri

Journals Department
Reviewer comments:

Reviewer #1 (Comments for the Author):

Safety and Feasibility Assessment of a Pharmacy-Driven AUC/MIC Vancomycin Dosing Protocol in a Multicenter Hospital System

Vancomycin is recommended first-line for the treatment of methicillin-resistant *Staphylococcus aureus* (MRSA) infections and the pharmacodynamic parameter most closely associated with efficacy is the AUC:MIC ratio. While vancomycin troughs were used as a surrogate for AUC:MIC for many years, more recent literature has suggested targeting troughs of 15-20 has led to excess toxicity. Because of this and the advent of calculators and programs able to calculate an AUC, the most recent

vancomycin guidelines recommend calculating an AUC when using vancomycin therapeutic drug monitoring. In this paper, the authors describe their transition to AUC-guided vancomycin dosing, comparing two time periods pre- and post-AUC implementation.

Major Comments

- The introduction is very lengthy compared to the rest of the paper, would suggest trimming down paragraphs 1-4 to streamline the paper read
- I appreciate your team reporting out on two different definitions of AKI and for pointing out pertinent limitations at the end of the manuscript
- The AKIN definition of AKI is increase in SCr {greater than or equal to} 0.3 mg/dL, SCr {greater than or equal to} 1.5 times baseline, or urine volume {less than or equal to} 0.5 mL/kg/hour for 6 hours. Since you didn't report on urine volume, you will have to define this as a modified AKIN definition and describe why urine output wasn't included (which you mention as a limitation, but should be defined in the methods)
- Similarly, the vancomycin guideline definition also includes "a decrease in calculated creatinine CL (CLcr) of 50% from baseline on 2 consecutive days," so your definition is modified from the guideline definition, or you will want to adjust to include this variable
- Methods: Can you describe a bit more about the intervention. How were vancomycin concentrations drawn in each of the study arms, did protocols in terms of when to draw concentrations change? Does pharmacy-driven entail all aspects of vancomycin, dosing and levels? When did AUC roll-out occur, right at the beginning of the post arm (4/1/21) or earlier to allow some time to acclimate to the switch? What Bayesian model was used (program name and version number)? What was the pharmacist vs. provider's roles in vancomycin dose adjustments? Quite a bit more in the methods on how this was implemented can be useful to help frame the results
- Methods paragraph 5: cost definition mentions length of hospital stay, but this isn't mentioned in the results
- Methods final paragraph: logistic regression analysis is mentioned at the end of methods, but then not reported in results
- Why did contrast rates go up so much from 2019 to 2021? Something potentially to add to your discussion. Also, was this group pre-emptively separated from the other concomitant nephrotoxins, and if so, why?
- Is there a way you could collect clinical outcomes outside of nephrotoxicity and LOS? It'd be great to have mortality, readmission, or infection recurrence data on this large of a cohort. I understand this may not be feasible, and is reflected in your paper's title (Safety and Feasibility)

Minor Comments

- Abstract background: capitalize Staphylococcus and un-italicize methicillin-resistant
- Abstract objectives: change wording to "fewer AKI events" or "lower AKI rates." Also would switch parenthesis to acute kidney injury (AKI)
- Abstract results: Add word concomitant before nephrotoxin if that is what is implied here since vancomycin itself is a nephrotoxin (also applies throughout the paper)
 - o Be clear which group is which when presenting %s (i.e. 7.6% vs. 9.4% AUC 7.6% vs. trough 9.4%)
 - o "there were 2.1 labs drawn per person..." I'd change your denominator here from patients to patient-days, since the duration of vancomycin will greatly impact the number of labs drawn
 - You'll also want to be clear what "labs" entails - is this just vancomycin concentrations? If so, I would state this, since "labs" could be chem7, etc.
 - o "The median total dose of vancomycin..." like the labs above, would normalize this to median daily dose of vancomycin. If duration was shortened, you can report that here too, but would separate dose and duration
- Introduction paragraph 1: no need to clarify intravenous formulation in parentheses
- Methods paragraph 3: spell out IRB
- Methods paragraph 6: did you collect the number of concomitant nephrotoxic agents? That would be interesting to report out, not just yes or no, but if number of nephrotoxins (1,2,3,4+) impacted AKI rates as part of your logistic regression
- Results paragraph 2: when giving percentage results, you will want both numerators and denominators, so readers don't need to guess which one it is (i.e. 13.9% (n=2507) 13.9% (349/2507))
- Results paragraph 3: same comments as the abstract, would normalize all results to patient-days (despite durations between groups being similar) - for dose, cost, labs
- Results paragraph 3: do you consider 0.2 labs/person clinically significant? If not, would include the word statistically and include the statistics.
 - o Also, this result does not match the abstract - results/table say 1.9 vs. 2.1 while the abstract says 2.1 vs. 2.3
- Results paragraph 3: I would combine cost outcomes as you define in your methods. Also, no mention of length of stay costs when that was mentioned in the methods?
- Discussion paragraph 1: you mention the short duration potentially being a cause of the lack of AKI difference between groups. What is the subgroup analysis of your results for say patients who received at least 5 days of vancomycin?
- Discussion paragraph 1, last couple of sentences: I'm confused how the critical care patients impacted your results? You excluded patients with "unstable renal function" per your methods, so these patients by definition didn't have unstable renal function
- Discussion paragraph 3, last line: rather than saying "If this study looked only at vancomycin durations greater than 4 days, there may not be any difference in lab draws.." why not just do that analysis and report out the outcome? For this and AKI rates this may be helpful as a subgroup analysis since you mention it twice in your discussion

- Table 1: you report out decimals for ICU admission and contrast, so I'd do the same for concomitant nephrotoxins to keep consistency
 - o How is first serum creatinine defined? Initial SCr on admission (as defined in methods) or first SCr after vanco initiation? Would re-word in table 1 to clarify
- Figure 1: would report out the breakdown of exclusion reasons in both arms in the figure, rather than just "8,943 patients excluded"

Reviewer #2 (Comments for the Author):

This is a well-written report describing safety and feasibility associated with implementation of a vancomycin AUC/MIC dosing protocol at a large hospital system, including several community hospitals. Please find some suggestions/comments/questions below:

Abstract:

Background, line 1: please capitalize all organize names (*S. aureus*)

Objective: Should the last line read "administered vancomycin dose during admission"?

Introduction: You set the stage well here with the history of recommendations and why you got to where you are today. While some of this could be streamlined, I think it reads well and provides excellent background.

Methodology:

Study endpoints, baseline characteristics: in the third line, consider revising to "If the patient received piperacillin-tazo..., they were considered to have received a nephrotoxic agent."

In this section (maybe under "Study sites"), it may be helpful to give more context to the sites. How big are the hospitals? How much vancomycin use at each (e.g. how many PK consults per day)? How many pharmacists to handle each hospital? Is vancomycin use an automatic consult, or does it have to be ordered by a prescriber? You mention that pharmacists liked the Bayesian software because it saved time, so this could be helpful to further set the stage.

Results: Paragraph 2 is duplicative to Table 2. You could simplify and add to paragraph 1 ("There was no difference in AKI between groups (Table 2), regardless of definition used.")

I would also be interested in the following results:

You mentioned missing data in the AUC group. Do you have an idea what the indications were on those that were included? It would be interesting to see, as you have noted, if this impacted your results overall.

Did the move to AUC/MIC dosing have any impact on improperly drawn labs? For example, did your rate of mistimed troughs, improperly drawn (e.g. not at steady state) troughs, the number extra that had to be drawn to correct improperly timed, etc. change with the implementation and movement to standard lab times?

Can you do a subgroup analysis of those who received more than 4 days of vancomycin? Did that impact your results?

Discussion:

One piece you could consider adding to the discussion would be a discussion of whether labs are even needed in this group. Per your statement, you may have missed some troughs because vanc was d/c'd before it got to steady state, and therefore was not drawn. Why was AUC drawn? Should it be if you know the course will only be 3.5 days? It might warrant a discussion on whether all patients should always get an immediate AUC calculated, or if there could be ways to adjust.

Figure 1: Please expand with numbers for exclusion criteria, if available. In the legend, you could remove the line that starts "there were 2507...".

Staff Comments:

Preparing Revision Guidelines

To submit your modified manuscript, log onto the eJP submission site at <https://spectrum.msubmit.net/cgi-bin/main.plex>. Go to Author Tasks and click the appropriate manuscript title to begin the revision process. The information that you entered when you first submitted the paper will be displayed. Please update the information as necessary. Here are a few examples of required

updates that authors must address:

Please return the manuscript within 60 days; if you cannot complete the modification within this time period, please contact me. If you do not wish to modify the manuscript and prefer to submit it to another journal, please notify me of your decision immediately so that the manuscript may be formally withdrawn from consideration by Microbiology Spectrum.

Spectrum Manuscript 03313-22

Safety and Feasibility Assessment of a Pharmacy-Driven AUC/MIC Vancomycin Dosing Protocol in a Multicenter Hospital System

Vancomycin is recommended first-line for the treatment of methicillin-resistant *Staphylococcus aureus* (MRSA) infections and the pharmacodynamic parameter most closely associated with efficacy is the AUC:MIC ratio. While vancomycin troughs were used as a surrogate for AUC:MIC for many years, more recent literature has suggested targeting troughs of 15-20 has led to excess toxicity. Because of this and the advent of calculators and programs able to calculate an AUC, the most recent vancomycin guidelines recommend calculating an AUC when using vancomycin therapeutic drug monitoring. In this paper, the authors describe their transition to AUC-guided vancomycin dosing, comparing two time periods pre- and post-AUC implementation.

Major Comments

- The introduction is very lengthy compared to the rest of the paper, would suggest trimming down paragraphs 1-4 to streamline the paper read
- I appreciate your team reporting out on two different definitions of AKI and for pointing out pertinent limitations at the end of the manuscript
- The AKIN definition of AKI is increase in SCr ≥ 0.3 mg/dL, SCr ≥ 1.5 times baseline, or urine volume ≤ 0.5 mL/kg/hour for 6 hours. Since you didn't report on urine volume, you will have to define this as a modified AKIN definition and describe why urine output wasn't included (which you mention as a limitation, but should be defined in the methods)
- Similarly, the vancomycin guideline definition also includes "a decrease in calculated creatinine CL (CLcr) of 50% from baseline on 2 consecutive days," so your definition is modified from the guideline definition, or you will want to adjust to include this variable
- Methods: Can you describe a bit more about the intervention. How were vancomycin concentrations drawn in each of the study arms, did protocols in terms of when to draw concentrations change? Does pharmacy-driven entail all aspects of vancomycin, dosing and levels? When did AUC roll-out occur, right at the beginning of the post arm (4/1/21) or earlier to allow some time to acclimate to the switch? What Bayesian model was used (program name and version number)? What was the pharmacist vs. provider's roles in vancomycin dose adjustments? Quite a bit more in the methods on how this was implemented can be useful to help frame the results
- Methods paragraph 5: cost definition mentions length of hospital stay, but this isn't mentioned in the results
- Methods final paragraph: logistic regression analysis is mentioned at the end of methods, but then not reported in results
- Why did contrast rates go up so much from 2019 to 2021? Something potentially to add to your discussion. Also, was this group pre-emptively separated from the other concomitant nephrotoxins, and if so, why?
- Is there a way you could collect clinical outcomes outside of nephrotoxicity and LOS? It'd be great to have mortality, readmission, or infection recurrence data on this large of a cohort. I understand this may not be feasible, and is reflected in your paper's title (Safety and Feasibility)

Minor Comments

- Abstract background: capitalize *Staphylococcus* and un-italicize methicillin-resistant
- Abstract objectives: change wording to “fewer AKI events” or “lower AKI rates.” Also would switch parenthesis to acute kidney injury (AKI)
- Abstract results: Add word concomitant before nephrotoxin if that is what is implied here since vancomycin itself is a nephrotoxin (also applies throughout the paper)
 - Be clear which group is which when presenting %s (i.e. 7.6% vs. 9.4% → AUC 7.6% vs. trough 9.4%)
 - “there were 2.1 labs drawn per person...” → I’d change your denominator here from patients to patient-days, since the duration of vancomycin will greatly impact the number of labs drawn
 - You’ll also want to be clear what “labs” entails – is this just vancomycin concentrations? If so, I would state this, since “labs” could be chem7, etc.
 - “The median total dose of vancomycin...” like the labs above, would normalize this to median daily dose of vancomycin. If duration was shortened, you can report that here too, but would separate dose and duration
- Introduction paragraph 1: no need to clarify intravenous formulation in parentheses
- Methods paragraph 3: spell out IRB
- Methods paragraph 6: did you collect the number of concomitant nephrotoxic agents? That would be interesting to report out, not just yes or no, but if number of nephrotoxins (1,2,3,4+) impacted AKI rates as part of your logistic regression
- Results paragraph 2: when giving percentage results, you will want both numerators and denominators, so readers don’t need to guess which one it is (i.e. 13.9% (n=2507) → 13.9% (349/2507))
- Results paragraph 3: same comments as the abstract, would normalize all results to patient-days (despite durations between groups being similar) – for dose, cost, labs
- Results paragraph 3: do you consider 0.2 labs/person clinically significant? If not, would include the word statistically and include the statistics.
 - Also, this result does not match the abstract – results/table say 1.9 vs. 2.1 while the abstract says 2.1 vs. 2.3
- Results paragraph 3: I would combine cost outcomes as you define in your methods. Also, no mention of length of stay costs when that was mentioned in the methods?
- Discussion paragraph 1: you mention the short duration potentially being a cause of the lack of AKI difference between groups. What is the subgroup analysis of your results for say patients who received at least 5 days of vancomycin?
- Discussion paragraph 1, last couple of sentences: I’m confused how the critical care patients impacted your results? You excluded patients with “unstable renal function” per your methods, so these patients by definition didn’t have unstable renal function
- Discussion paragraph 3, last line: rather than saying “If this study looked only at vancomycin durations greater than 4 days, there may not be any difference in lab draws...” why not just do that analysis and report out the outcome? For this and AKI rates this may be helpful as a subgroup analysis since you mention it twice in your discussion
- Table 1: you report out decimals for ICU admission and contrast, so I’d do the same for concomitant nephrotoxins to keep consistency
 - How is first serum creatinine defined? Initial SCr on admission (as defined in methods) or first SCr after vanco initiation? Would re-word in table 1 to clarify
- Figure 1: would report out the breakdown of exclusion reasons in both arms in the figure, rather than just “8,943 patients excluded”

Response to Reviewers

Major Comments

- The introduction is very lengthy compared to the rest of the paper, would suggest trimming down paragraphs 1-4 to streamline the paper read
 - **I cut down several parts of the intro to make it more concise as denote by my track changes.**
- I appreciate your team reporting out on two different definitions of AKI and for pointing out pertinent limitations at the end of the manuscript
- The AKIN definition of AKI is increase in SCr ≥ 0.3 mg/dL, SCr ≥ 1.5 times baseline, or urine volume ≤ 0.5 mL/kg/hour for 6 hours. Since you didn't report on urine volume, you will have to define this as a modified AKIN definition and describe why urine output wasn't included (which you mention as a limitation, but should be defined in the methods)
 - **See study endpoints under primary endpoint – I clarified the definition we used to reflect what our data reporting software could “pull” for us**
 - **Also, nursing documentation is lacking so we did think this would be a very feasible measure to include**
- Similarly, the vancomycin guideline definition also includes “a decrease in calculated creatinine CL (CLcr) of 50% from baseline on 2 consecutive days,” so your definition is modified from the guideline definition, or you will want to adjust to include this variable
 - **See study endpoints under primary endpoint – I clarified the definition we used to reflect what our data reporting software could “pull” for us**
- Methods: Can you describe a bit more about the intervention. How were vancomycin concentrations drawn in each of the study arms, did protocols in terms of when to draw concentrations change? Does pharmacy-driven entail all aspects of vancomycin, dosing and levels? When did AUC roll-out occur, right at the beginning of the post arm (4/1/21) or earlier to allow some time to acclimate to the switch? What Bayesian model was used (program name and version number)? What was the pharmacist vs. provider's roles in vancomycin dose adjustments? Quite a bit more in the methods on how this was implemented can be useful to help frame the results
 - **I included a new paragraph in the methods section explaining the roll-out and implementation of our software.**
- Methods paragraph 5: cost definition mentions length of hospital stay, but this isn't mentioned in the results
 - **This was removed. Unfortunately, this parameter was not the easiest to obtain (there are too many factors that would go into estimating the cost per person)**
- Methods final paragraph: logistic regression analysis is mentioned at the end of methods, but then not reported in results
 - **This sentence was deleted. We originally were planning on having this incorporated but decided not to include it as access to the statistician changed with no ability to run the logistic regression.**
- Why did contrast rates go up so much from 2019 to 2021? Something potentially to add to your discussion. Also, was this group pre-emptively separated from the other concomitant nephrotoxins, and if so, why?
 - **It's unclear why this was the case. I included additional wording in the manuscript to address.**
- Is there a way you could collect clinical outcomes outside of nephrotoxicity and LOS? It'd be great to have mortality, readmission, or infection recurrence data on this large of a cohort. I understand this may not be feasible, and is reflected in your paper's title (Safety and Feasibility)
 - **Addressed in conclusion and included some additional wording to state this could be performed in the future.**

Minor Comments

- Abstract background: capitalize Staphylococcus and un-italicize methicillin-resistant
 - **Done**
- Abstract objectives: change wording to “fewer AKI events” or “lower AKI rates.” Also would switch parenthesis to acute kidney injury (AKI)
 - **Done**

- Abstract results: Add word concomitant before nephrotoxin if that is what is implied here since vancomycin itself is a nephrotoxin (also applies throughout the paper)
 - Done
- Be clear which group is which when presenting %s (i.e. 7.6% vs. 9.4% → AUC 7.6% vs. trough 9.4%)
 - Done
- “there were 2.1 labs drawn per person...” → I’d change your denominator here from patients to patient-days, since the duration of vancomycin will greatly impact the number of labs drawn
 - To clarify, would you recommend the denominator as vancomycin patient days? Not hospital patient days? Ex: (2.1 labs/person)/(3.6 days of vancomycin) → 0.58 labs/person-days
- You’ll also want to be clear what “labs” entails – is this just vancomycin concentrations? If so, I would state this, since “labs” could be chem7, etc.
 - Done
- “The median total dose of vancomycin...” like the labs above, would normalize this to median daily dose of vancomycin. If duration was shortened, you can report that here too, but would separate dose and duration
 - In order to do this would you like the statistics to be rerun? As a second-year pharmacy resident now at a new institution, I do not have easy access to my data anymore. If we would like to pursue this I can reach out to the pharmacist who ran our statistical analysis.
- Introduction paragraph 1: no need to clarify intravenous formulation in parentheses
 - Done
- Methods paragraph 3: spell out IRB
 - Done
- Methods paragraph 6: did you collect the number of concomitant nephrotoxic agents? That would be interesting to report out, not just yes or no, but if number of nephrotoxins (1,2,3,4+) Impacted AKI rates as part of your logistic regression
 - Unfortunately, we did not collect this measure. We did not have the ability to perform logistic regression either.
- Results paragraph 2: when giving percentage results, you will want both numerators and denominators, so readers don’t need to guess which one it is (i.e. 13.9% (n=2507) → 13.9% (349/2507))
 - Done
- Results paragraph 3: same comments as the abstract, would normalize all results to patient-days (despite durations between groups being similar) – for dose, cost, labs
 - See above
- Results paragraph 3: do you consider 0.2 labs/person clinically significant? If not, would include the word statistically and include the statistics.
 - Done
- Also, this result does not match the abstract – results/table say 1.9 vs. 2.1 while the abstract says 2.1 vs. 2.3
 - Done
- Results paragraph 3: I would combine cost outcomes as you define in your methods. Also, no mention of length of stay costs when that was mentioned in the methods?
 - Done
- Discussion paragraph 1: you mention the short duration potentially being a cause of the lack of AKI difference between groups. What is the subgroup analysis of your results for say patients who received at least 5 days of vancomycin?
 - The goal of our study was to focus on the practical use of vancomycin in the hospital system. I agree that this would be an ideal measure to report on. I do think this study paves the way for further studies looking at more specific indications and clinical outcomes (e.g., only looking at patients who received vancomycin greater than 5 days).
- Discussion paragraph 1, last couple of sentences: I’m confused how the critical care patients impacted your results? You excluded patients with “unstable renal function” per your methods, so these patients by definition didn’t have unstable renal function
 - At the start of vancomycin, patients would have been excluded if they had unstable renal function (which we particularly see with critical care patients). I do think it’s important to mention that included critical care

patients could impact the results because they are more likely to have fluctuating renal function at any point in their stay. While the included critical care patients did not have unstable renal function initially, their high risk status could make them more likely to have an AKI in the subsequent days while on vancomycin therapy.

- Discussion paragraph 3, last line: rather than saying “If this study looked only at vancomycin durations greater than 4 days, there may not be any difference in lab draws..” why not just do that analysis and report out the outcome? For this and AKI rates this may be helpful as a subgroup analysis since you mention it twice in your discussion
 - See the above referring to discussion, paragraph 1
- Table 1: you report out decimals for ICU admission and contrast, so I’d do the same for concomitant nephrotoxins to keep consistency
 - Done
- How is first serum creatinine defined? Initial SCr on admission (as defined in methods) or first SCr after vanco initiation? Would re-word in table 1 to clarify
 - Done
- Figure 1: would report out the breakdown of exclusion reasons in both arms in the figure, rather than just “8,943 patients excluded”
 - Done

December 20, 2022

Dr. Morgan Simonian Phillips
Sentara RMH Medical Center
2010 Health Campus Drive
Harrisonburg, Virginia 22801

Re: Spectrum03313-22R1 (Safety and Feasibility Assessment of a Pharmacy-Driven AUC/MIC Vancomycin Dosing Protocol in a Multicenter Hospital System)

Dear Dr. Morgan Simonian Phillips:

Thank you for submitting your manuscript to Microbiology Spectrum. Responses to comments of reviewer 2 are missing. When submitting the revised version of your paper, please provide (1) point-by-point responses to the issues raised by the reviewers as file type "Response to Reviewers," not in your cover letter, and (2) a PDF file that indicates the changes from the original submission (by highlighting or underlining the changes) as file type "Marked Up Manuscript - For Review Only". Please use this link to submit your revised manuscript - we strongly recommend that you submit your paper within the next 60 days or reach out to me. Detailed instructions on submitting your revised paper are below.

Link Not Available

Sincerely,

Tulip Jhaveri

Journals Department
Reviewer comments:

Staff Comments:

Preparing Revision Guidelines

Please return the manuscript within 60 days; if you cannot complete the modification within this time period, please contact me. If you do not wish to modify the manuscript and prefer to submit it to another journal, please notify me of your decision immediately so that the manuscript may be formally withdrawn from consideration by Microbiology Spectrum.

Response to Reviewers

Reviewer 1

Major Comments

- The introduction is very lengthy compared to the rest of the paper, would suggest trimming down paragraphs 1-4 to streamline the paper read
 - **I cut down several parts of the intro to make it more concise as denote by my track changes.**
- I appreciate your team reporting out on two different definitions of AKI and for pointing out pertinent limitations at the end of the manuscript
- The AKIN definition of AKI is increase in SCr ≥ 0.3 mg/dL, SCr ≥ 1.5 times baseline, or urine volume ≤ 0.5 mL/kg/hour for 6 hours. Since you didn't report on urine volume, you will have to define this as a modified AKIN definition and describe why urine output wasn't included (which you mention as a limitation, but should be defined in the methods)
 - **See study endpoints under primary endpoint – I clarified the definition we used to reflect what our data reporting software could “pull” for us**
 - **Also, nursing documentation is lacking so we did think this would be a very feasible measure to include**
- Similarly, the vancomycin guideline definition also includes “a decrease in calculated creatinine CL (CLcr) of 50% from baseline on 2 consecutive days,” so your definition is modified from the guideline definition, or you will want to adjust to include this variable
 - **See study endpoints under primary endpoint – I clarified the definition we used to reflect what our data reporting software could “pull” for us**
- Methods: Can you describe a bit more about the intervention. How were vancomycin concentrations drawn in each of the study arms, did protocols in terms of when to draw concentrations change? Does pharmacy-driven entail all aspects of vancomycin, dosing and levels? When did AUC roll-out occur, right at the beginning of the post arm (4/1/21) or earlier to allow some time to acclimate to the switch? What Bayesian model was used (program name and version number)? What was the pharmacist vs. provider's roles in vancomycin dose adjustments? Quite a bit more in the methods on how this was implemented can be useful to help frame the results
 - **I included a new paragraph in the methods section explaining the roll-out and implementation of our software.**
- Methods paragraph 5: cost definition mentions length of hospital stay, but this isn't mentioned in the results
 - **This was removed. Unfortunately, this parameter was not the easiest to obtain (there are too many factors that would go into estimating the cost per person)**
- Methods final paragraph: logistic regression analysis is mentioned at the end of methods, but then not reported in results
 - **This sentence was deleted. We originally were planning on having this incorporated but decided not to include it as access to the statistician changed with no ability to run the logistic regression.**
- Why did contrast rates go up so much from 2019 to 2021? Something potentially to add to your discussion. Also, was this group pre-emptively separated from the other concomitant nephrotoxins, and if so, why?
 - **It's unclear why this was the case. I included additional wording in the manuscript to address.**
- Is there a way you could collect clinical outcomes outside of nephrotoxicity and LOS? It'd be great to have mortality, readmission, or infection recurrence data on this large of a cohort. I understand this may not be feasible, and is reflected in your paper's title (Safety and Feasibility)
 - **Addressed in conclusion and included some additional wording to state this could be performed in the future.**

Minor Comments

- Abstract background: capitalize Staphylococcus and un-italicize methicillin-resistant
 - **Done**
- Abstract objectives: change wording to “fewer AKI events” or “lower AKI rates.” Also would switch parenthesis to acute kidney injury (AKI)
 - **Done**

- Abstract results: Add word concomitant before nephrotoxin if that is what is implied here since vancomycin itself is a nephrotoxin (also applies throughout the paper)
 - Done
- Be clear which group is which when presenting %s (i.e. 7.6% vs. 9.4% → AUC 7.6% vs. trough 9.4%)
 - Done
- “there were 2.1 labs drawn per person...” → I’d change your denominator here from patients to patient-days, since the duration of vancomycin will greatly impact the number of labs drawn
 - To clarify, would you recommend the denominator as vancomycin patient days? Not hospital patient days? Ex: (2.1 labs/person)/(3.6 days of vancomycin) → 0.58 labs/person-days
- You’ll also want to be clear what “labs” entails – is this just vancomycin concentrations? If so, I would state this, since “labs” could be chem7, etc.
 - Done
- “The median total dose of vancomycin...” like the labs above, would normalize this to median daily dose of vancomycin. If duration was shortened, you can report that here too, but would separate dose and duration
 - In order to do this would you like the statistics to be rerun? As a second-year pharmacy resident now at a new institution, I do not have easy access to my data anymore. If we would like to pursue this I can reach out to the pharmacist who ran our statistical analysis.
- Introduction paragraph 1: no need to clarify intravenous formulation in parentheses
 - Done
- Methods paragraph 3: spell out IRB
 - Done
- Methods paragraph 6: did you collect the number of concomitant nephrotoxic agents? That would be interesting to report out, not just yes or no, but if number of nephrotoxins (1,2,3,4+) Impacted AKI rates as part of your logistic regression
 - Unfortunately, we did not collect this measure. We did not have the ability to perform logistic regression either.
- Results paragraph 2: when giving percentage results, you will want both numerators and denominators, so readers don’t need to guess which one it is (i.e. 13.9% (n=2507) → 13.9% (349/2507))
 - Done
- Results paragraph 3: same comments as the abstract, would normalize all results to patient-days (despite durations between groups being similar) – for dose, cost, labs
 - See above
- Results paragraph 3: do you consider 0.2 labs/person clinically significant? If not, would include the word statistically and include the statistics.
 - Done
- Also, this result does not match the abstract – results/table say 1.9 vs. 2.1 while the abstract says 2.1 vs. 2.3
 - Done
- Results paragraph 3: I would combine cost outcomes as you define in your methods. Also, no mention of length of stay costs when that was mentioned in the methods?
 - Done
- Discussion paragraph 1: you mention the short duration potentially being a cause of the lack of AKI difference between groups. What is the subgroup analysis of your results for say patients who received at least 5 days of vancomycin?
 - The goal of our study was to focus on the practical use of vancomycin in the hospital system. I agree that this would be an ideal measure to report on. I do think this study paves the way for further studies looking at more specific indications and clinical outcomes (e.g., only looking at patients who received vancomycin greater than 5 days).
- Discussion paragraph 1, last couple of sentences: I’m confused how the critical care patients impacted your results? You excluded patients with “unstable renal function” per your methods, so these patients by definition didn’t have unstable renal function
 - At the start of vancomycin, patients would have been excluded if they had unstable renal function (which we particularly see with critical care patients). I do think it’s important to mention that included critical care

patients could impact the results because they are more likely to have fluctuating renal function at any point in their stay. While the included critical care patients did not have unstable renal function initially, their high risk status could make them more likely to have an AKI in the subsequent days while on vancomycin therapy.

- Discussion paragraph 3, last line: rather than saying "If this study looked only at vancomycin durations greater than 4 days, there may not be any difference in lab draws.." why not just do that analysis and report out the outcome? For this and AKI rates this may be helpful as a subgroup analysis since you mention it twice in your discussion
 - See the above referring to discussion, paragraph 1
- Table 1: you report out decimals for ICU admission and contrast, so I'd do the same for concomitant nephrotoxins to keep consistency
 - Done
- How is first serum creatinine defined? Initial SCr on admission (as defined in methods) or first SCr after vanco initiation? Would re-word in table 1 to clarify
 - Done
- Figure 1: would report out the breakdown of exclusion reasons in both arms in the figure, rather than just "8,943 patients excluded"
 - Done

Reviewer 2

Abstract:

Background, line 1: please capitalize all organize names (S. aureus)

- Done

Objective: Should the last line read "administered vancomycin dose during admission"?

- Done

Introduction: You set the stage well here with the history of recommendations and why you got to where you are today. While some of this could be streamlined, I think it reads well and provides excellent background.

- Thank you, I cut down the intro as per the above comments from reviewer 1

Methodology:

Study endpoints, baseline characteristics: in the third line, consider revising to "If the patient received piperacillin-tazo..., they were considered to have received a nephrotoxic agent."

- Done

In this section (maybe under "Study sites"), it may be helpful to give more context to the sites. How big are the hospitals? How much vancomycin use at each (e.g. how many PK consults per day)? How many pharmacists to handle each hospital? Is vancomycin use an automatic consult, or does it have to be ordered by a prescriber? You mention that pharmacists liked the Bayesian software because it saved time, so this could be helpful to further set the stage.

- Done

Results: Paragraph 2 is duplicative to Table 2. You could simplify and add to paragraph 1 ("There was no difference in AKI between groups (Table 2), regardless of definition used.")

- Done

I would also be interested in the following results:

You mentioned missing data in the AUC group. Do you have an idea what the indications were on those that were included? It would be interesting to see, as you have noted, if this impacted your results overall.

- Unfortunately, I think because of transitioning to the new documentation formatting, most of the indications were unavailable.

Did the move to AUC/MIC dosing have any impact on improperly drawn labs? For example, did your rate of mistimed troughs, improperly drawn (e.g. not at steady state) troughs, the number extra that had to be drawn to correct improperly timed, etc. change with the implementation and movement to standard lab times?

- With the new AUC-dosing protocol, the timing of vancomycin levels usually becomes irrelevant (generally recommended to draw at least 2 hours post vancomycin-infusion of the first dose after a loading dose is given). This is not a point we addressed.

Can you do a subgroup analysis of those who received more than 4 days of vancomycin? Did that impact your results?

- I addressed this above in reviewer 1's comments

Discussion:

One piece you could consider adding to the discussion would be a discussion of whether labs are even needed in this group. Per your statement, you may have missed some troughs because vanc was d/c'd before it got to steady state, and therefore was not drawn. Why was AUC drawn? Should it be if you know the course will only be 3.5 days? It might warrant a discussion on whether all patients should always get an immediate AUC calculated, or if there could be ways to adjust.

- I agree with this and this is something we have discussed implementing in follow-up education. We think our pharmacists could limit number of vancomycin levels obtained to help further reduce costs. I included a statement about this in the last discussion paragraph.

Figure 1: Please expand with numbers for exclusion criteria, if available. In the legend, you could remove the line that starts "there were 2507...".

- Addressed as per reviewer 1's comment

December 24, 2022

Dr. Morgan Simonian Phillips
Sentara RMH Medical Center
2010 Health Campus Drive
Harrisonburg, Virginia 22801

Re: Spectrum03313-22R2 (Safety and Feasibility Assessment of a Pharmacy-Driven AUC/MIC Vancomycin Dosing Protocol in a Multicenter Hospital System)

Dear Dr. Morgan Simonian Phillips:

Thank you for submitting your manuscript to Microbiology Spectrum. Major revisions are needed before I can consider this manuscript to move to the next stage. I would highly recommend the author to add line numbers for the text. When you respond to reviewer's comments, direct the readers to the line numbers for which these changes have been made. It is very hard for the reviewers and editors to track these changes you've made without any clear direction.

For example, reviewer 1 commented:

"Abstract background: capitalize Staphylococcus and un-italicize methicillin-resistant"

Your response was: Done

Editor: Instead of saying done, would encourage the author to respond something along the lines of:

We have modified the sentence to reflect the following in Line ***:

"Vancomycin is used for Gram-positive infections, including methicillin-resistant Staphylococcus aureus."

Please do this for all the places you have responded with 'done.' Only then, can we consider this manuscript for further review.

Additionally, here are some further edits I'd recommend:

- Importance: Please change 'I' to 'we' as I believe there are more than one authors.
- Abstract Background: Would keep the first sentence as is "Vancomycin is used for Gram-positive infections, including methicillin-resistant Staphylococcus aureus."
- Methods - under Vancomycin AUC/MIC Pharmacy Dosing protocol - please change the tense of the first paragraph to past tense.
- Discussion - There was no identifiable reason 'for' higher contrast use in the AUC group (please add 'for')
- Re: the author's question "To clarify, would you recommend the denominator as vancomycin patient days? Not hospital patient days? Ex. (2.1 labs/person)/(3.6 days of vancomycin) 0.58 labs/person-days"
Editor: Yes, we mean Vancomycin patient days, please change it to Vancomycin patient days
- Re: the author's question "In order to do this would you like the statistics to be rerun? As a secondyear pharmacy resident now at a new institution, I do not have easy access to my data anymore. If we would like to pursue this I can reach out to the pharmacist who ran our statistical analysis."
Editor: No problem, that's fine
- Results paragraph 2: when giving percentage results, you will want both numerators and denominators, so readers don't need to guess which one it is (i.e. 13.9% (n=2507) → 13.9% (349/2507)). Despite the table being there, this (including numerator/denominator) needs to be mentioned in the text and not deleted.
- Results paragraph 3: same comments as the abstract, would normalize all results to patient-days (despite durations between groups being similar) - for dose, cost, labs. Your response was See above. This needs to be clarified further.

Link Not Available

ASM policy requires that data be available to the public upon online posting of the article, so please verify all links to sequence records, if present, and make sure that each number retrieves the full record of the data. If a new accession number is not linked or a link is broken, provide production staff with the correct URL for the record. If the accession numbers for new data are not publicly accessible before the expected online posting of the article, publication of your article may be delayed; please contact

the ASM production staff immediately with the expected release date.

Sincerely,

Tulip Jhaveri

Journals Department
Reviewer comments:

Staff Comments:

Preparing Revision Guidelines

Please return the manuscript within 60 days; if you cannot complete the modification within this time period, please contact me. If you do not wish to modify the manuscript and prefer to submit it to another journal, please notify me of your decision immediately so that the manuscript may be formally withdrawn from consideration by Microbiology Spectrum.

Response to Reviewers

Reviewer 1

Major Comments

The introduction is very lengthy compared to the rest of the paper, would suggest trimming down paragraphs 1-4 to streamline the paper read

We address the intro to make it more concise as referenced in lines 68-96

I appreciate your team reporting out on two different definitions of AKI and for pointing out pertinent limitations at the end of the manuscript

The AKIN definition of AKI is increase in SCr ≥ 0.3 mg/dL, SCr ≥ 1.5 times baseline, or urine volume ≤ 0.5 mL/kg/hour for 6 hours. Since you didn't report on urine volume, you will have to define this as a modified AKIN definition and describe why urine output wasn't included (which you mention as a limitation, but should be defined in the methods)

We have modified the sentence to reflect the following in Lines 140-143

Similarly, the vancomycin guideline definition also includes "a decrease in calculated creatinine CL (CLcr) of 50% from baseline on 2 consecutive days," so your definition is modified from the guideline definition, or you will want to adjust to include this variable

We have modified the sentence to reflect the following in Lines 144-147

Methods: Can you describe a bit more about the intervention. How were vancomycin concentrations drawn in each of the study arms, did protocols in terms of when to draw concentrations change? Does pharmacy-driven entail all aspects of vancomycin, dosing and levels? When did AUC roll-out occur, right at the beginning of the post arm (4/1/21) or earlier to allow some time to acclimate to the switch? What Bayesian model was used (program name and version number)? What was the pharmacist vs. provider's roles in vancomycin dose adjustments? Quite a bit more in the methods on how this was implemented can be useful to help frame the results

We have modified the sentence to reflect the following in Lines 124-136

Methods paragraph 5: cost definition mentions length of hospital stay, but this isn't mentioned in the results

This was removed. Unfortunately, this parameter was not the easiest to obtain (there are too many factors that would go into estimating the cost per person)

Methods final paragraph: logistic regression analysis is mentioned at the end of methods, but then not reported in results

This sentence was deleted. We originally were planning on having this incorporated but decided not to include it as access to the statistician changed with no ability to run the logistic regression.

Why did contrast rates go up so much from 2019 to 2021? Something potentially to add to your discussion. Also, was this group pre-emptively separated from the other concomitant nephrotoxins, and if so, why?

We have modified the sentence to reflect the following in Lines 193-195

Is there a way you could collect clinical outcomes outside of nephrotoxicity and LOS? It'd be great to have mortality, readmission, or infection recurrence data on this large of a cohort. I understand this may not be feasible, and is reflected in your paper's title (Safety and Feasibility)

We have modified the sentence to reflect the following in Lines 243-246

Minor Comments

Abstract background: capitalize Staphylococcus and un-italicize methicillin-resistant

We have modified the sentence to reflect the following in Line 13

Abstract objectives: change wording to “fewer AKI events” or “lower AKI rates.” Also would switch parenthesis to acute kidney injury (AKI)

We have modified the sentence to reflect the following in Line 17

Abstract results: Add word concomitant before nephrotoxin if that is what is implied here since vancomycin itself is a nephrotoxin (also applies throughout the paper)

We have modified the sentence to reflect the following in Line 156

Be clear which group is which when presenting %s (i.e. 7.6% vs. 9.4% → AUC 7.6% vs. trough 9.4%)

We have modified the sentence to reflect the following in Lines 168-170

“there were 2.1 labs drawn per person...” → I’d change your denominator here from patients to patient-days, since the duration of vancomycin will greatly impact the number of labs drawn

We updated the values to reflect vancomycin patient-days in “Table 3”

You’ll also want to be clear what “labs” entails – is this just vancomycin concentrations? If so, I would state this, since “labs” could be chem7, etc.

We have modified the sentence to reflect the following in Lines 18, 25, 60, 181

“The median total dose of vancomycin...” like the labs above, would normalize this to median daily dose of vancomycin. If duration was shortened, you can report that here too, but would separate dose and duration

We updated the values to reflect vancomycin patient-days in “Table 3”

Introduction paragraph 1: no need to clarify intravenous formulation in parentheses

We have modified the sentence to reflect the following in Line 70

Methods paragraph 3: spell out IRB

We have modified the sentence to reflect the following in Line 122

Methods paragraph 6: did you collect the number of concomitant nephrotoxic agents? That would be interesting to report out, not just yes or no, but if number of nephrotoxins (1,2,3,4+) impacted AKI rates as part of your logistic regression

Unfortunately, we did not collect this measure. We did not have the ability to perform logistic regression either.

Results paragraph 2: when giving percentage results, you will want both numerators and denominators, so readers don’t need to guess which one it is (i.e. 13.9% (n=2507) → 13.9% (349/2507))

We have modified the sentence to reflect the following in Lines 23-26; 166-170; 176-177.

Results paragraph 3: same comments as the abstract, would normalize all results to patient-days (despite durations between groups being similar) – for dose, cost, labs

For word count purposes, we excluded labs per person in the abstract (Lines 26-27)

Results paragraph 3: do you consider 0.2 labs/person clinically significant? If not, would include the word statistically and include the statistics.

We have modified the sentence to reflect the following in Line 182-184

Also, this result does not match the abstract – results/table say 1.9 vs. 2.1 while the abstract says 2.1 vs. 2.3

These results from abstract were removed for word count purposes (lines 26-27)

Results paragraph 3: I would combine cost outcomes as you define in your methods. Also, no mention of length of stay costs when that was mentioned in the methods?

We have modified the sentence to reflect the following in Lines 181-182

Discussion paragraph 1: you mention the short duration potentially being a cause of the lack of AKI difference between groups. What is the subgroup analysis of your results for say patients who received at least 5 days of vancomycin?

The goal of our study was to focus on the practical use of vancomycin in the hospital system. I agree that this would be an ideal measure to report on. I do think this study paves the way for further studies looking at more specific indications and clinical outcomes (e.g., only looking at patients who received vancomycin greater than 5 days).

Discussion paragraph 1, last couple of sentences: I'm confused how the critical care patients impacted your results? You excluded patients with "unstable renal function" per your methods, so these patients by definition didn't have unstable renal function

At the start of vancomycin, patients would have been excluded if they had unstable renal function (which we particularly see with critical care patients). I do think it's important to mention that included critical care patients could impact the results because they are more likely to have fluctuating renal function at any point in their stay. While the included critical care patients did not have unstable renal function initially, their high risk status could make them more likely to have an AKI in the subsequent days while on vancomycin therapy.

Discussion paragraph 3, last line: rather than saying "If this study looked only at vancomycin durations greater than 4 days, there may not be any difference in lab draws.." why not just do that analysis and report out the outcome? For this and AKI rates this may be helpful as a subgroup analysis since you mention it twice in your discussion

I agree that this would be an ideal measure to report on. I do think this study paves the way for further studies looking at more specific indications and clinical outcomes (e.g., only looking at patients who received vancomycin greater than 5 days).

Table 1: you report out decimals for ICU admission and contrast, so I'd do the same for concomitant nephrotoxins to keep consistency

This has been updated in "Table 1: Received contrast dye"

How is first serum creatinine defined? Initial SCr on admission (as defined in methods) or first SCr after vanco initiation? Would re-word in table 1 to clarify

This has been updated in "Table 1: Baseline serum creatinine"

Figure 1: would report out the breakdown of exclusion reasons in both arms in the figure, rather than just "8,943 patients excluded

The full exclusion criteria has been updated as referenced in "Figure 1"

Reviewer 2

Abstract:

Background, line 1: please capitalize all organize names (S. aureus)

We have modified the sentence to reflect the following in Line 13

Objective: Should the last line read "administered vancomycin dose during admission"?

We have modified the sentence to reflect the following in Line 18

Introduction: You set the stage well here with the history of recommendations and why you got to where you are today. While some of this could be streamlined, I think it reads well and provides excellent background.

Thank you, we cut down the introduction significantly.

Methodology:

Study endpoints, baseline characteristics: in the third line, consider revising to "If the patient received piperacillin-tazo...., they were considered to have received a nephrotoxic agent."

We have modified the sentence to reflect the following in Line 156

In this section (maybe under "Study sites"), it may be helpful to give more context to the sites. How big are the hospitals? How much vancomycin use at each (e.g. how many PK consults per day)? How many pharmacists to handle each hospital? Is vancomycin use an automatic consult, or does it have to be ordered by a prescriber? You mention that pharmacists liked the Bayesian software because it saved time, so this could be helpful to further set the stage.

We have modified the sentence to reflect the following in Lines 128-129

Results: Paragraph 2 is duplicative to Table 2. You could simplify and add to paragraph 1 ("There was no difference in AKI between groups (Table 2), regardless of definition used.")

We have modified the sentence to reflect the following in Line 172

I would also be interested in the following results:

You mentioned missing data in the AUC group. Do you have an idea what the indications were on those that were included? It would be interesting to see, as you have noted, if this impacted your results overall.

Unfortunately, I think because of transitioning to the new documentation formatting, most of the indications were unavailable.

Did the move to AUC/MIC dosing have any impact on improperly drawn labs? For example, did your rate of mistimed troughs, improperly drawn (e.g. not at steady state) troughs, the number extra that had to be drawn to correct improperly timed, etc. change with the implementation and movement to standard lab times?

With the new AUC-dosing protocol, the timing of vancomycin levels usually becomes irrelevant (generally recommended to draw at least 2 hours post vancomycin-infusion of the first dose after a loading dose is given). This is not a point we addressed.

Can you do a subgroup analysis of those who received more than 4 days of vancomycin? Did that impact your results?

I addressed this above in reviewer 1's comments

Discussion:

One piece you could consider adding to the discussion would be a discussion of whether labs are even needed in this group. Per your statement, you may have missed some troughs because vanc was d/c'd before it got to steady state, and therefore was not drawn. Why was AUC drawn? Should it be if you know the course will only be 3.5 days? It might warrant a discussion on whether all patients should always get an immediate AUC calculated, or if there could be ways to adjust.

I agree with this and this is something we have discussed implementing in follow-up education. We think our pharmacists could limit number of vancomycin levels obtained to help further reduce costs. I included a statement about this in the last discussion paragraph.

Figure 1: Please expand with numbers for exclusion criteria, if available. In the legend, you could remove the line that starts "there were 2507...".

The goal of our study was to focus on the practical use of vancomycin in the hospital system. I agree that this would be an ideal measure to report on. I do think this study paves the way for further studies looking at more specific indications and clinical outcomes (e.g., only looking at patients who received vancomycin greater than 5 days).

Additionally, here are some further edits I'd recommend:

Importance: Please change 'I' to 'we' as I believe there are more than one authors.

We have modified the sentence to reflect the following in Line 59

Abstract Background: Would keep the first sentence as is "Vancomycin is used for Gram-positive infections, including methicillin-resistant Staphylococcus aureus."

We have modified the sentence to reflect the following in Line 13

Methods - under Vancomycin AUC/MIC Pharmacy Dosing protocol - please change the tense of the first paragraph to past tense.

We have modified the sentence to reflect the following in Lines 125-132

Discussion - There was no identifiable reason 'for' higher contrast use in the AUC group (please add 'for')

We have modified the sentence to reflect the following in Line 194

January 19, 2023

Dr. Morgan Simonian Phillips
Sentara RMH Medical Center
2010 Health Campus Drive
Harrisonburg, Virginia 22801

Re: Spectrum03313-22R3 (Safety and Feasibility Assessment of a Pharmacy-Driven AUC/MIC Vancomycin Dosing Protocol in a Multicenter Hospital System)

Dear Dr. Morgan Simonian Phillips:

Your manuscript has been accepted, and I am forwarding it to the ASM Journals Department for publication. You will be notified when your proofs are ready to be viewed.

Sincerely,

Tulip Jhaveri
Editor, Microbiology Spectrum
